# LongBondEliminator: A Molecular Simulation Tool to Remove Ring Penetrations in Biomolecular Simulation Systems

**DOI:** 10.3390/biom13010107

**Published:** 2023-01-05

**Authors:** Daipayan Sarkar, Martin Kulke, Josh V. Vermaas

**Affiliations:** MSU-DOE Plant Research Laboratory and Department of Biochemistry and Molecular Biology, Michigan State University, East Lansing, MI 48824, USA

**Keywords:** molecular simulation, protein glycosylation, ring penetration, minimization, lignin, virus

## Abstract

We develop a workflow, implemented as a plugin to the molecular visualization program VMD, that can fix ring penetrations with minimal user input. LongBondEliminator, detects ring piercing artifacts by the long, strained bonds that are the local minimum energy conformation during minimization for some assembled simulation system. The LongBondEliminator tool then automatically treats regions near these long bonds using multiple biases applied through NAMD. By combining biases implemented through the collective variables module, density-based forces, and alchemical techniques in NAMD, LongBondEliminator will iteratively alleviate long bonds found within molecular simulation systems. Through three concrete examples with increasing complexity, a lignin polymer, an viral capsid assembly, and a large, highly glycosylated protein aggrecan, we demonstrate the utility for this method in eliminating ring penetrations from classical MD simulation systems. The tool is available via gitlab as a VMD plugin, and has been developed to be generically useful across a variety of biomolecular simulations.

## 1. Introduction

Particle-based molecular simulation, either through classical molecular dynamics (MD) [1,2], quantum techniques [3,4], or emerging machine learned force field approaches [5,6], requires precise knowledge of the positions of each particle prior to simulation. For simple single component systems, this is largely a solved problem, as structural databases [7] or recent artificial intelligence-powered structure prediction tools [8,9] deliver precise atomic coordinates for many systems of interest. However, real biological systems are more complex, with multiple components in close proximity within a crowded environment [10,11]. For example, glycans added to viral proteins are essential for their function [12], as demonstrated by recent simulations of the SARS-CoV-2 spike protein [13], but are only rarely captured in crystallography [14]. New tools are expanding the limit of what is possible through molecular simulation, both in biological complexity and size [15,16,17,18,19].

While developing these tools to add these essential molecular details to complement predicted and experimentally determined structures are crucial to creating accurate molecular models for biological phenomena, they can lead to new problems as well. As new elements are added to a system, there is the potential for newly placed elements to occupy the same space within the structure, creating unphysical structures that cannot be minimized away [20,21,22]. While overlapping structural elements can be dealt with by hand, or through integrating environment-aware placement algorithms into existing molecular simulation toolkits, these solutions typically scale poorly with system complexity or size and create substantial overhead on the part of the user or the programmer.

Of particular concern are ring-piercing artifacts, where newly placed entities lead to a situation where two nominally bonded atoms are on opposite sides of a small ring [20]. Unlike larger rings that can occur through polymerization [23,24], bonds through small rings produce unphysical configurations that are a problem regardless of the simulation method. In quantum mechanical or machine learning approaches, where the bonding topology is defined by geometry, the bond will be split across the ring, creating an alternative topology. For classical systems, a pierced ring configuration (e.g., Figure 1) is a local minima, as extending the bond well past its equilibrium value is lower in energy than creating even momentarily unfavorable Lennard-Jones interactions [20]. Although the typical solution of treating the whole system alchemically temporarily and minimizing can fix many systems where atomic overlap is subtle, more vigorous methods may be required. For instance, recently developed toolkits for mixing lipids use carving potentials to make space to add a lipid and may move atoms with long bonds randomly [25]. CHARMM-GUI also has a protocol for detecting ring penetrations [26], and can automatically resolve some ring penetrations caused by sterols [27].

By adapting existing tools and features within the molecular simulation engine NAMD [1], we develop a workflow, implemented as a plugin within VMD [28], that can fix ring penetrations with minimal user input. The tool, LongBondEliminator (LBE), detects ring piercing artifacts by the long, strained bonds that are the local minimum energy conformation during minimization [20,22], and is available via gitlab (https://gitlab.msu.edu/vermaaslab/LongBondElminiator, accessed on 31 December 2021) or Zenodo [29]. The regions near these long bonds are then treated using the collective variables module [30], density-based forces [31], and alchemical techniques combined to attempt to alleviate the long bond. Through three concrete examples with increasing complexity, a lignin polymer, a viral capsid assembly, and a large, highly glycosylated protein, we demonstrate the utility of this method in eliminating ring-piercing artifacts from classical MD simulation systems.

## 2. Materials and Methods

### 2.1. LongBondEliminator Minimization Algorithm

The implementation to LBE is derived empirically, and has its origins in solving ring piercing artifacts when building lignin polymers [22]. Lignin is a polyaromatic compound found in the secondary cell wall of plants, and is synthesized through radical coupling reactions [32]. Based on this synthesis, lignin structure is highly heterogeneous [32]. When the full polymer was unfurled from optimized dimers, it was possible that a bond might be placed through the aromatic ring, and a mechanism was required to fix the placement. What worked for LigninBuilder [22] was to search for bonds longer than 1.65 Å in a given minimized structure, which would indicate stretched bonds that likely pierce a ring. This is analogous to the detection method from Whitmore et al. [20], which explicitly looked for extended bonds as a symptom of ring penetration. With a few changes, this basic workflow became LBE (Figure 2), a general-purpose tool that handles a wide variety of ring piercing artifacts across a diverse array of initial configurations. To eliminate long bonds that pierce a ring, LBE uses multiple techniques. Long bonds are treated by alchemical free energy perturbation (FEP) methods, and the overlapping elements are pushed away from one another through a combination of GridForces [33] to push on atoms according to a defined density, and the colvars (collective variables) module [30] to push on atoms based on user-defined criteria within NAMD [1].

The 1.65 Å distance cutoff originally implemented in LigninBuilder was slightly longer than the equilibrium bond length between carbons, oxygens, and hydrogens within lignin. However, as the methodology was applied to other biological systems with greater coverage of the periodic table, this cutoff would flag normal bonds, particularly to large elements such as sulfur, as being unusually long. LBE now uses a flexible cutoff depending on the equilibrium bond lengths found in the simulation parameter files for the CHARMM force field. As shown in Figure 2, minimization stages will continue until the longest bond in the system, when minimized without flagged alchemical atoms, is less than 0.4 Å longer than the equilibrium bond length. However, to avoid infinite loops, the run will terminate after 100 minimization stages of 500 steps each. If no ring piercings are present in the initial structure and no bonds are extended, this requirement will be met after the first step, and represents the minimum path through Figure 2.

However, if stretched bonds are found after minimization, where the actual bond length is more than 0.4 Å larger than the equilibrium bond length specified by the force field, multiple biasing methods are applied to resolve the long bonds. The overall goal is to displace the atoms involved in the long bond and their immediate neighbors away from one another. In LigninBuilder, where there are relatively few pierced rings that lead to a long bond, it was sufficient to create a repulsive density and collective variable near the location of the long bond, and to treat the nearby atoms alchemically so that the nonbonded interaction between atoms does not increase to infinity at short interatomic distances [22]. In effect, the alchemical treatment exploits soft core potentials used during alchemical simulations to soften the overall energy landscape used during minimization. While these approaches have been broadly successful, and are retained in LBE, they are insufficient alone to solve tangled rings in highly glycosylated structures.

In addition to the procedures above, we have added new tricks to LBE in order to improve the structures that emerge for complex systems. We now focus on the worst bonds initially, selecting long bonds that are up to 0.1 Å of the most stretched bonds in the system in a given iteration through the workflow. Prior iterations of LBE would attempt to solve all long bonds at once, which proved to be unstable for particularly complex piercing artifacts. We also find that creating an additional bias to simply move the two atoms that constitute a long bond perpendicular to the bond vector connecting them can unstick stubborn geometries. This is implemented through a collective variable that attempts to increase the distance between the center of masses between the long bond and nearby atoms to up to 10 Å. To further contribute to the unsticking phenomena, the force constants are doubled if the same bond remains the longest between successive iterations for the overall workflow. Since the forces applied to individual atoms in the system can be quite large, the default is for all potential chiral centers within the molecule to be constrained using the extrabonds feature of NAMD [1], largely by repeating logic found in the chirality checking and modification scripts already present in VMD [28] to accommodate chiral centers in carbohydrates. Together, this methodology successfully solves ring piercings across multiple systems with minimal user input, as exemplified by Figure 3.

### 2.2. Test Cases

We have prepared three demonstrations that highlight the effectiveness of LBE over a range of different systems. We feature a small lignin polymer created by LigninBuilder [22], a facet of the viral envelope for the AstraZeneca COVID-19 vaccine (ChAdOx1 AZD1222/Vaxzevria), which also needed a helping hand to resolve ring piercing artifacts [34], and a highly glycosylated protein aggrecan. The construction for each system is described in turn in the following subsections. Since LBE explicitly modifies a NAMD configuration file, all test cases were run using NAMD 2.14 [1] without hydrating water molecules. Multiple minimizations were orchestrated in VMD [28] using a Tcl script, which we provide within the gitlab repository.

Although the LongBondElminator can also work in solvated systems, the test cases were run without solvent to minimize computational expense so that they could be easily replicated by a potential user with minimal computational cost. As a result, we used a dielectric of 80 to create weaker electrostatic interactions, as might be the case in a solvated system. Since we are operating in a vacuum, long-range electrostatics via PME was of minimal perceived value to run the test cases, and so the systems are not minimized within a periodic unit cell.

#### 2.2.1. Lignin Polymer

The chosen lignin polymer is the 66${}^{\mathrm{th}}$ polymer from the spruce database from previous lignin polymer libraries [35]. The polymer is relatively large for a lignin, with 5322 atoms together accounting for 40.5 kDa of mass, but is the smallest example system under study here (Figure 4). LigninBuilder [22] inflated the topology proscribed into an initial structure, which featured multiple ring penetrations. For minimization, we use the CHARMM force field for lignin [36], with 12 Å nonbonded cutoffs.

#### 2.2.2. Virus

LBE can also be applied to large biomolecular systems, such as viral capsid assemblies, to reduce the clashes between adjacent asymmetric units. Algorithms often based on shape-complimentary between docking pairs can often lead to large atom clashes and ring piercings in biomolecular assemblies. Docking algorithms based on rigid body transformations are one such example, where a great extent of pairwise atom clashes and pierced aromatic rings can be initially generated as part of a bioassembly. Using docking to establish protein-protein interaction is computationally inexpensive. However, rigid body docking or fitting algorithms often do not account for any underlying potential energy surface governing the interaction of individual biomolecular subunits, and often need further refinement. To test the protocol we present here, as a second test case we revisit a recently published viral capsid structure. A single facet from the biological assembly for the viral vector ChAdOx1, which is adapted from chimpanzee adenovirus Y25 (ChAd-Y25), forms the basis for the AstraZeneca COVID-19 vaccine (AZD1222/Vaxzevria) [34] (see Figure 5), and is approximately 4 million atoms large even before solvation. As described in the primary literature [34], the original facet models determined by rigid-body fits by assembling multiple proteins together featured ring penetration artifacts. LBE is the evolution for the original detangling methods previously presented. During minimization with LBE, interactions within protein components are treated using the CHARMM36m force field [37,38]. Further refinement for the cryo-electron microscopy data once the long bonds are eliminated are not considered here, but would be subsequent steps after the LBE process completes.

#### 2.2.3. Glycosylated Protein

Another test case is the highly glycosylated aggrecan protein, highlighting the effectiveness for LBE in this use case. The Alphafold 2 predicted structure of aggrecan core protein (amino acids 17-2530, uniprot id P16112 [41]) was manually elongated in VMD [28] by rotating the protein segments parallel to the x-axis. The structure was minimized and further elongated with steered molecular dynamics simulations in vacuum applying a constant force of $100\;\mathrm{kcal}\;{}^{-2}$ between all three globular domain pairings G1–G3 for 6 ps (Figure 6). Afterwards, the structure was relaxed with equilibrium molecular dynamics simulations in vacuum for 6 ns. Because of the large amount of glycosylations in aggrecan, the carbohydrate chains were attached automatically to the relaxed core protein structure with a VMD script utilizing psfgen functionality. Shorter keratan sulfate chains were attached to 9 amino acids in the interglobular domain (IGD), while longer chains were added in the keratan sulfate-rich domain (KSD) to 30 amino acids. Long chondroitin sulfate chains were added to 159 amino acids in the chondroitin sulfate rich domains (CSD1 and CSD2). The resulting glycsolylated aggrecan contained 309,247 atoms totaling a mass of 2756 kDa. The initial arbitrary rotation of the amino acid side chains that the carbohydrate chains were attached to, resulted in carbohydrate chains tangled with each other and the protein backbone leading to over 800 hexose ring piercings, especially in CSD1 (Figure 6). LBE procedure minimized this structure using the CHARMM36m force field for proteins [37,38] and sugars [42,43,44] and 12 Å nonbonded cutoffs.

### 2.3. User Guide

The longbondeliminator.tcl file defines multiple functions, leveraging features available within VMD as appropriate. Only the first three of these functions listed below are intended for general use, while the others implement pieces of the LBE algorithm.

minimize <namdconfigurationfile> <deletelines> <namdrunargs> <cutoff> <genchirality> <writeintermediates>, which goes through the logic to eliminate long bonds. This function is responsible for parsing the NAMD configuration file, and loading in the molecular simulation system that the NAMD configuration file refers to.namdconfigurationfile refers to a NAMD configuration file that minimizes the system, but reveals ring penetrations when executed. The existing simulation file will be retained during the subsequent minimization procedures, with the exception of the last few lines.deletelines refers to the number of lines that are deleted from the end of the NAMD configuration file. The assumption is that the last lines in the NAMD configuration file will minimize and run a simulation, which LBE will replace. 3 lines are the default.namdrunargs shows how you might run a NAMD simulation on your system, e.g., “namd2 +p16” would tell NAMD to use 16 processors when running the simulation. The default is “namd2”.cutoff is the metric used to determine when to terminate. Concretely, when the most stretched bond is less than the cutoff larger than the equilibrium bond distance for the interaction found within the force field, we assume the system to be minimized. The default is “0.4”, with units in Å.genchirality is either a 0 or a 1. If 1, LBE will generate improper dihedral constraints that are meant to maintain the chirality of potential chiral centers within the molecule. The default option is to generate the constraints.writeintermediates is also either a 0 or a 1. If 1, LBE will write a binary coordinate file for the end state at each step. The default option is not to write these intermediate files.getlargestdeviation <molid> <parameterlist> <filename>, which measures the most stretched bond in a molecular simulation system prepared in VMD.molid is the molecule identification number for a molecule loaded into VMD that you would like to measure.parameterlist is a Tcl list that contains all the parameter files necessary to simulate this molecular model.filename is the name of the file where the output should be written.tagdeviation <molid> <parameterlist> will store the sum of large deviations (>0.1 Å) into the user field for an atom at a particular frame from within a VMD trajectory.runloop is the function that implements the central runloop shown in Figure 2, and is intended only to be called from the minimize function.forcelongestbond is the function that creates a target location 10 Å away from the center of the bond vector between a stretched bond, and does so perfectly perpendicular to the bond vector.writenamdconf writes a new NAMD configuration file based on the existing NAMD configuration file.genchiralityextrabondsfile generates the extrabonds file that maintains chirality of the initial model system.minimizationscriptname determines what the new NAMD configuration file should be called.buildbondtable builds a table of bond parameters from a list of filenames.loadsystem loads in the structure and coordinates listed in the original NAMD configuration file.parseinputfile parses the NAMD configuration file for key parameters needed to implement the LBE algorithm.

Examples of how LBE was run to generate the results are available in test.tcl. Note that LBE can either be loaded into VMD as needed by sourcing longbondeliminator.tcl, or by adding the location to autopath and then using the usual ‘package require’ semantics.

### 2.4. Analysis

The intermediate and end states for the test cases were analyzed to assess the geometry present in the final structure. The longest bond after a given minimization step in Figure 2 was recorded using getlargestdeviation, with timestep 0 representing the original input structure. We also developed a visualization for the minimization process that utilizes tagdeviation, adding the degree of bond extension to individual atoms for coloration. To minimize the number of atomselections that needed to be made, only bonds that were extended by at least 0.1 Å were added to the user field within VMD for visualization on a per frame basis.

For the virus capsid testcase, where the structure was amenable to analysis via MolProbity [45], we track the intermediate MolProbity and clashscores over the course of the minimization. The individual intermediate structures for the virus capsid were analyzed using the MolProbity tools implemented within PHENIX [46]. Like the long bond analysis, the results were plotted in Python, leveraging numpy [47] and matplotlib [48] libraries.

## 3. Results

LBE is focused with the extent to which a bond is stretched and uses this as the primary metric to assess convergence. The length of the longest bond over multiple cycles through the workflow diagram (Figure 2) is provided in Figure 7. We find that for the three test cases chosen here, only 20 cycles are required to arrive at a structure where the bond stretches are below 0.4 Å. At this degree of stretch, the energy stored within the bond may still represent tens of kcal mol${}^{-1}$. This energy could conceivably be dissipated through further minimization steps or equilibration, and thus some thought on the part of the user is envisioned as to what the next steps should be. However, since a 0.4 Å bond stretch is inconsistent with a ring piercing event, it is conceivable that the final structure written in lbe-opt.coor is sufficiently robust to begin simulation.

Figure 7 highlights a common occurrence during minimization, namely that the longest bond goes through cycles of increasing and decreasing. The origin of this movement lies in the alchemical approach taken to allow close contact between atoms along the left branch of Figure 2, which facilitates motions of the stretched bond past the ring it is piercing. The soft-core potential also allows the stretched bond to relax and come closer to its equilibrium length, and may even fall below the cutoff. If the end state is below the final cutoff, the workflow traverses the right branch of Figure 2, reverting to standard Lennard-Jones potentials. If a ring remains pierced after the cycle is complete, the piercing bond will stretch again to an extended length and the machinery within the LongBondElminator will have another opportunity to fix the problem.

Figure 7 features another common trend, in that the number of steps that must be taken scales with the number of penetrations of aromatic rings present in the simulation system. Because LBE targets the longest bonds first, some initial ring penetrations may not be targeted initially. Only on successive steps are these new longest bonds targeted, until all stretch or deviations of the bonds are below the chosen cutoff, which defaults to 0.4 Å.

Given time, so far LBE has always found a configuration where the structures no longer feature ring piercings or ring penetrations. LBE clearly reduces the longest bonds indicative of ring penetrations. Whereas Figure 3 demonstrated local improvements to ring penetration artifacts, Figure 8 highlights progress at a global scale across the molecular system, as the spheres representing the extended bonds transition to cooler colors, indicating fewer bonds that are stretched going from the initial to the final state. However, Figure 8 also shows that there are still modestly stretched bonds in the final state, showing blue spheres where the bonds are still extended by more than 0.1 Å. At this level of extension, the bonds are close enough to equilibrium that the forces on individual atoms are not large enough to cause simulation instability on their own, and may be used as real simulation inputs.

As LBE applies alchemical transformations to potentially poor initial structures, it is of interest to track the quality of intermediate structure at every step of the protocol (Figure 2). Among the examples presented in this work, the ChAdOx1 virus capsid (PDB: 7RD1) was solved by integrating high-resolution cryo-EM experiments with all-atom molecular dynamics simulations, using an integrative modeling technique popularly known as Molecular Dynamics Flexible Fitting (MDFF) [31,49]. Since the initial structure was assembled from multiple subcomponents that were individually minimized, it was possible for hidden clashes to exist.

Figure 9 calculates the clash and MolProbity scores [45,50] for the different asymmetric units in a facet of an adenovirus with T25 point symmetry. Crucially, going from step 0, the initial structure, to step 1, after unperturbed minimization, the ring penetration artifacts present but hidden in the initial structure raise both the clash and MolProbity scores. The clash score, which is a subcomponent of the overall MolProbity score that also considers metrics such as bond lengths, rotamers and Ramachandran outliers, is typically improved by simple minimization for well behaved structures that do not feature ring penetrations. MolProbity scores can be thought of as resolution-equivalents, and thus lower scores correspond to higher quality structures. While both the clashscore and the overall MolProbity score initially increase as LBE does its work to resolve ring penetration artifacts, subsequent steps reduce atomic clashes within the structure and generally improve structure quality.

## 4. Discussion

The ideas behind LBE are not original, and instead the innovation is in packaging these ideas into a single workflow that can be tested or adapted to a given system of interest. For instance, the idea of transforming the atoms involved in long bonds stems from recommendations for GROMACS users, who find that using alchemical treatments for bad atomic contacts facilitates minimization for systems that would overflow single-precision force calculations. Similarly, LBE uses soft-core potentials to temporarily reduce the cost of bringing two atoms together. Utilizing biases implemented using collective variables or density grids are common tools to guide systems along productive pathways [30,51,52]. LBE takes these approaches and packages them into a single tool that can be readily integrated into structure building workflows by integrating into VMD [28].

We view LBE as eliminating a significant bottleneck in structure preparation for molecular simulation. Although most simulation systems do not feature ring penetrations or piercings, as single proteins or well resolved complexes are common simulation targets. For systems that feature ring penetrations, solving these unphysical artifacts is a substantial impediment to scientific progress. In some cases, the system is immediately unstable, and the MD integrator will stop the computation. The stretched bonds would naturally have a higher energy than an alternative structure without the unphysical geometry, and this can sometimes trigger failsafes within MD software that recognize unstable systems. However, the authors are aware of instances where entanglements or ring penetrations were stable during simulation and were only revealed after analysis [21], as a pierced ring remains a local minimum in the energy landscape for the system. Immediate system instability thus represents a best-case scenario, so that simulation dynamics is not perturbed, and compute time is used effectively. With LBE, it is possible to quickly verify, without significant scripting effort, if there are any long bonds present in the simulation system. If long bonds indicative of ring penetrations are found, LBE has a generalized workflow that can solve artifacts identified in various systems (Figure 8).

These advantages do not come without tradeoffs that an end-user should be aware of. We cannot claim that LBE is the most computationally efficient method for solving ring penetrations. LBE was designed to optimize against user time, delivering a protocol that works for diverse systems. As a consequence of the multiple NAMD features used beyond standard minimization, larger multi-million systems such as the ChAdOx1 vector may take a few hours of minimization cycles on a desktop to resolve all of the ring penetrations in the system. For a point of comparison on the real computational cost, all three examples were run sequentially overnight using a 16-core AMD Ryzen 9 5950X CPU. Thus, LBE is readily runnable on a desktop, rather than being confined to supercomputers. However, other methods developed for resolving ring penetrations between lipids and proteins are comparatively simpler, and certainly faster for many systems. CHARMM-GUI [26,27] and Membrane Mixer [25] take advantage of fast insertion of lipid or sterol into bilayers to either let dynamics reinsert a molecule after pulling it out or carve a space using density grids before adding a lipid in the first place. Alternatively, it may be possible to use robotic motion planning algorithms [53] or molecular dynamics with adaptive decision making [54] to engineer realistic molecular geometries for complex systems without the expensive minimization and reminimization steps.

Another potential algorithm would be to simply translate the atoms involved in long bonds perpendicular to the bond vector, similarly to the applied force. While reasonably fast to implement, in our testing such an approach was highly likely to flip stereocenters and be unsuitable for general use. For glycosylated systems, where stereochemistry dictates molecular properties and interactions, these flipped chiralities would introduce further unphysical geometries, which would be much harder to detect upon casual visual inspection of the minimized structure. Without chirality constraints, it is possible that LBE will also flip a stereocenter, just as the underlying density force methods are known to do. Thus, we strongly recommend running LBE in the default mode where all potential chiral centers are restrained, or independently adding these restraints to the minimization configuration script passed along to the LBE workflow.

In addition to the performance of LBE for large biomolecular systems, it must be noted that we recommend that LBE may only be the first step for a larger structural pipeline with data-guided molecular modeling approaches. For example, to solve the ChAdOx1 virus capsid structure (PDB:7RD1), the asymmetric unit after LBE treatment needed multiple subsequent refinement steps. The refinement stages combined cryo-EM map enhancement methods and MDFF with symmetry restraints [55] together with protein ensemble refinement using multiple cryo-EM density maps [51]. Manual interactive refinement using ISOLDE [56] was also performed, yielding the final structure [34]. These additional steps are beyond the scope of LBE, but are essential for developing the best model possible.

In summary, as biomolecular simulation systems become more complex and unwieldy, featuring multiple biopolymers and components, it is inevitable that more ring penetrations will be found during system construction. If not caught in advance, these ring penetrations would produce unphysical geometries (Figure 1), and thus should be handled systematically to resolve these ring penetrations, ideally without significant user input. The LBE approach, available from gitlab (https://gitlab.msu.edu/vermaaslab/LongBondElminiator, accessed on 31 December 2021) or Zenodo [29], implements a VMD plugin that has been demonstrated to be effective in solving ring piercing artifacts in multiple complex systems. Therefore, LBE represents a robust tool that could be considered the first line of defense to identify and fix these artifacts within molecular simulation systems.

## Figures and Tables

**Figure 1 biomolecules-13-00107-f001:**
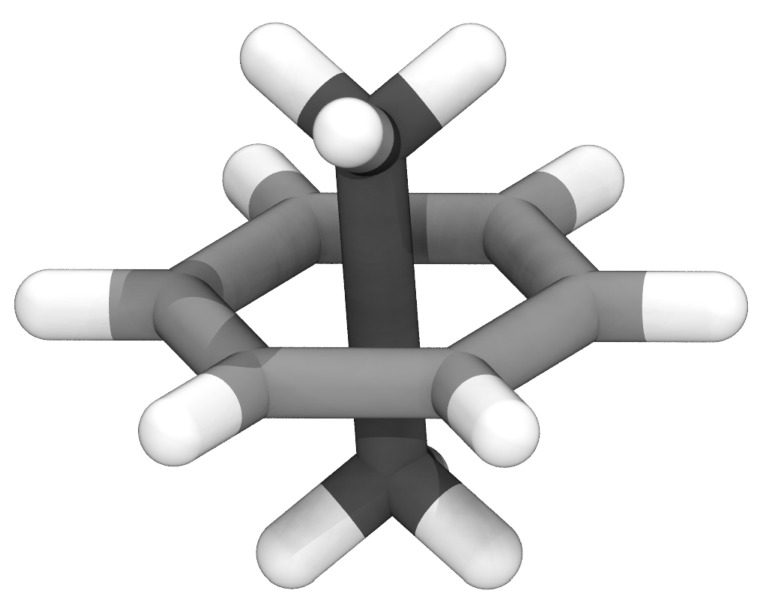
Example structure for ethane (darker carbons) piercing benzene (lighter carbons), highlighting the unphysical geometry with the extended carbon-carbon bond that is a local minimum in classical MD force fields.

**Figure 2 biomolecules-13-00107-f002:**
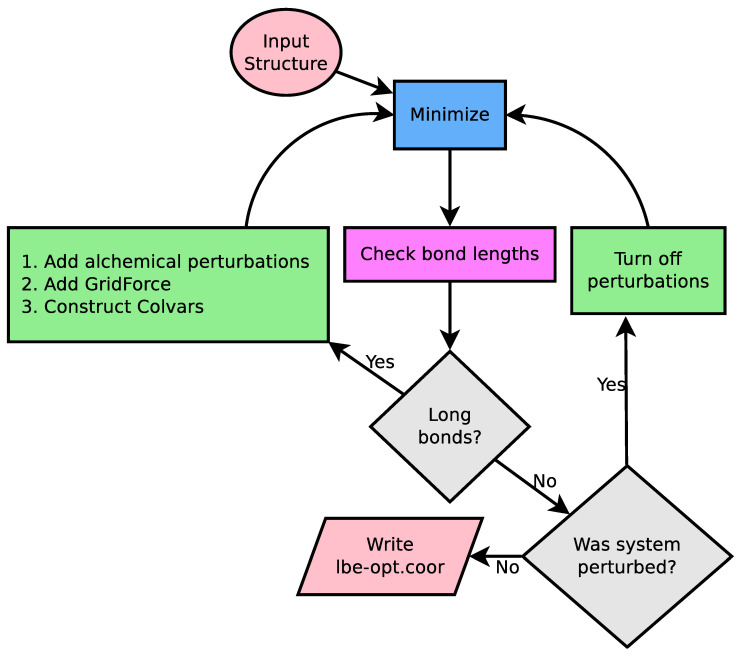
Logical flow diagram guiding the cycle for iterative minimization. A working NAMD configuration file is essentially the only input, providing an input structure, coordinates, and parameters. NAMD is used for minimization (blue square), and the primary analysis computes the bond lengths from simulation (pink square). The flow control is handled by the gray diamonds, which first compares the computed bond length against the equilibrium bond length in the simulation force field, and if no bonds are detected, asks if the previous simulation run had alchemical perturbations turned on. Modifications to the original NAMD configuration file are highlighted in green. If long bonds are detected, multiple features are added to the NAMD configuration file to relax the structure. Otherwise, if the previous simulation was perturbed, another minimization round is carried out without alchemical perturbation features enabled. When the procedure completes, we write a set of output coordinates.

**Figure 3 biomolecules-13-00107-f003:**
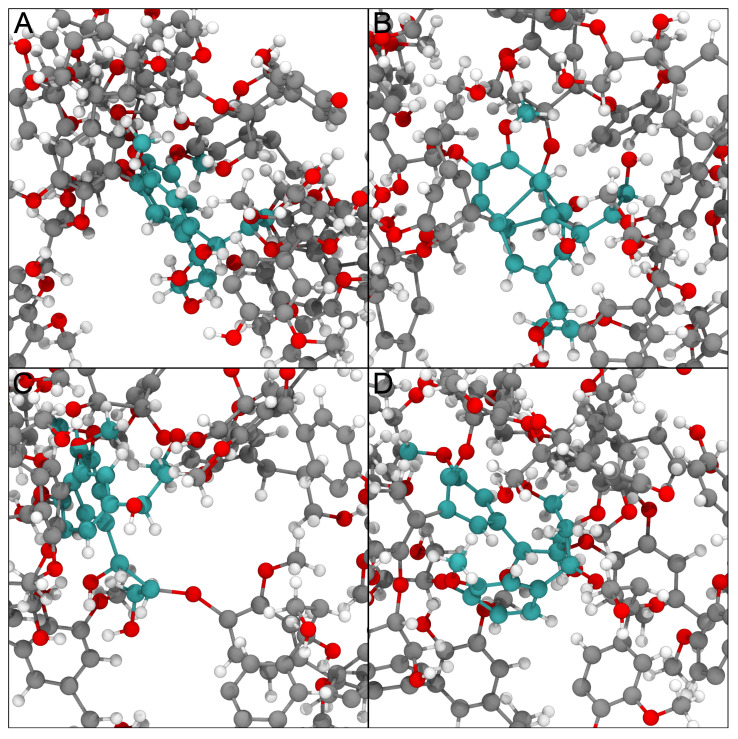
Example of how physically impossible structures are ameliorated by the LBE process. Different panels here represent snapshots taken along the LBE process for a single unphysical geometry in the lignin testcase. The original structure shown in panel (**A**) has two residues with cyan carbon atoms whose aromatic rings that are in close physical proximity based on the initial construction process. After minimization, these rings pierce one another, as shown in panel (**B**). By applying multiple biasing approaches, the rings are moved substantially, resulting in panel (**C**). After further minimization, the rest of the molecular environment moves the two residues of interest close to their original positions, but critically these residues are no longer overlapping one another. The structural representation in panel (**D**) is thus ready for further study.

**Figure 4 biomolecules-13-00107-f004:**
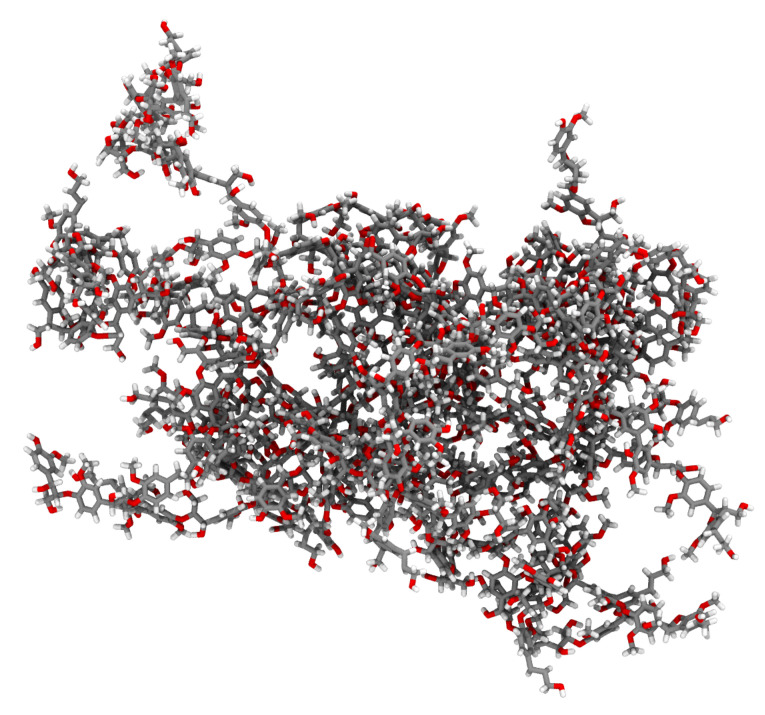
Initial structure for our example lignin polymer, featuring multiple aromatic ring piercings that must be resolved. In the shown representation, atoms are color coded, with gray for carbon, red for oxygen, and white for hydrogen.

**Figure 5 biomolecules-13-00107-f005:**
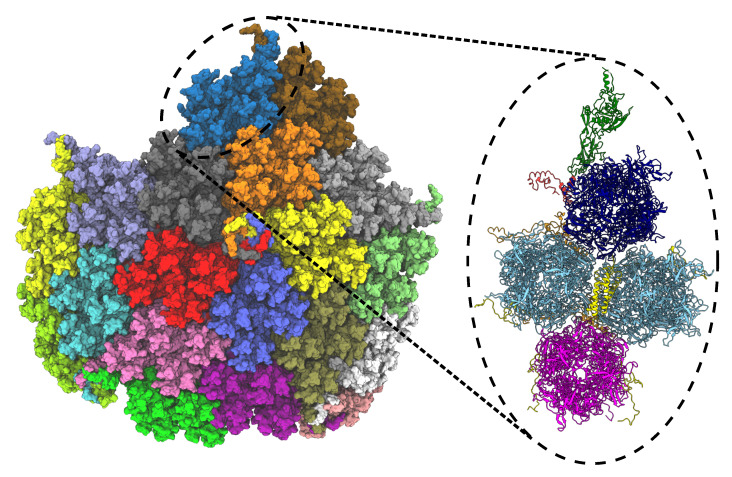
Facet of ChAdOx1 adapted from chimpanzee adenovirus Y25 (ChAd-Y25), which is the basis for the AstraZenica COVID-19 vaccine (AZD1222/Vaxzevria). The full capsid (T = 25) point symmetry of the viral vector consists of 60 copies of the asymmetric unit, which once optimized was published as PDB 7RD1 [34]. The full facet (left) comprises 19 asymmetric units, each represented by a surface representation with a unique color. The full facet illustration is prepared in VMD 1.9.4a55 [28]. The dashed circle (black) represents an asymmetric unit, which is expanded in the left panel to highlight the different proteins in the asymmetric unit of ChAdOx1 virus capsid. The unit contains a penton monomer (green), a trimeric peripentonal hexon (blue), two copies of 2${}^{\prime }$ hexons (sky blue), a copy of 3${}^{\prime }$ hexon (magenta), a tetrameric helical bundle protein IX (yellow), protein IIIa (red) and pre-hexon-linking protein VIII (orange). The illustration is prepared using ChimeraX [39,40].

**Figure 6 biomolecules-13-00107-f006:**
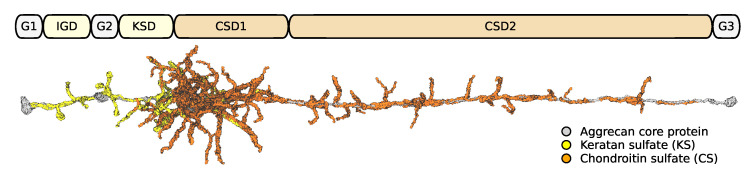
Structure of glycosylated aggrecan core protein showing the different domains. All domains with keratan sulfate or chondroitin sulfate glycosylations feature occasional ring piercings through the hexose carbohydrate ring.

**Figure 7 biomolecules-13-00107-f007:**
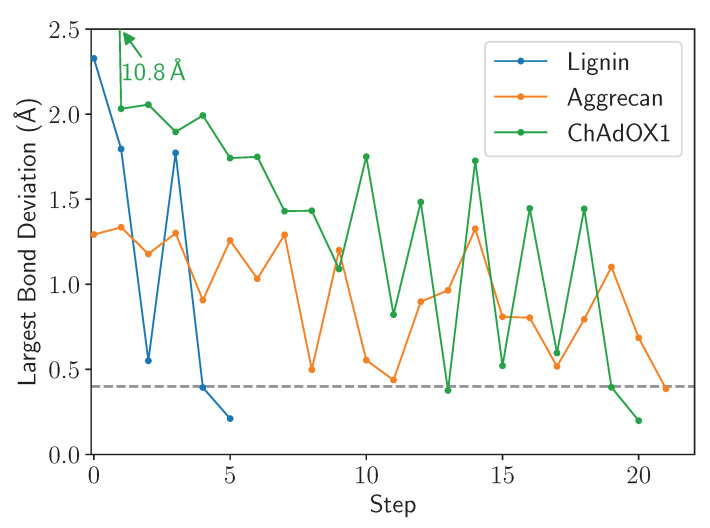
Example traces highlighting the eventual convergence for all structures after repeated cycles of the LBE procedure. In this instance, step 0 is the initial structure, prior to minimization, while the subsequent steps follow the primary workflow outlined in Figure 2. The 0.4 Å convergence criteria is highlighted by a gray dashed line.

**Figure 8 biomolecules-13-00107-f008:**
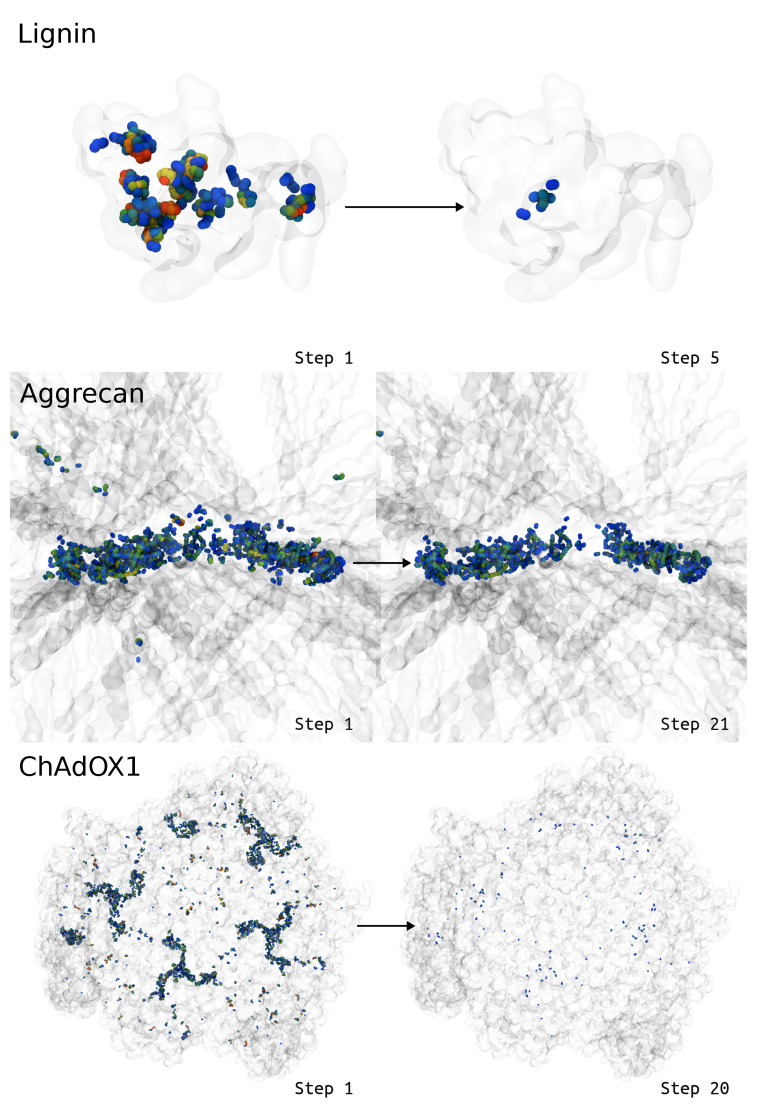
Visual representations for how the deviations change in the context of the larger biomolecule (transparent ghost), with atoms that demonstrate large bond deviations drawn as spheres and colored according to the sum of the bond deviations around a given atom. Blue colors represent small bond deviations, while redder colors represent larger bond deviations beyond 1 Å. Note that for aggrecan, we zoom into the region with the greatest glycosylation density, as this was the most challenging region to deal with ring piercing interactions. Animations for these visualizations are available as Appendix A.

**Figure 9 biomolecules-13-00107-f009:**
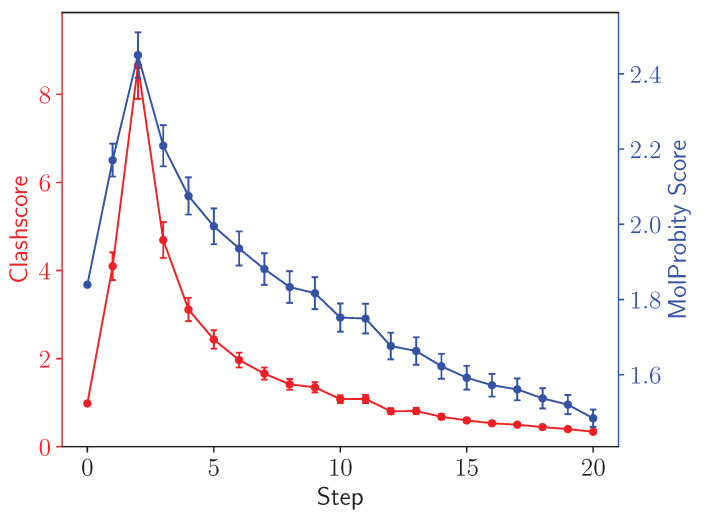
Clash (red, left axis) and MolProbity (blue, right axis) scores [50] for asymmetric units in the ChAdOx1 virus capsid. Scores are computed for individual asymmetric units within the facet and then averaged, with the uncertainties representing the standard error computed from among the different asymmetric units. The clashscore and the overall MolProbity score increase after the initial step as the long bond traverses out of a pierced aromatic ring.

## Data Availability

The LongBondEliminator plugin, in addition to the inputs and outputs used to develop this analysis, are available from gitlab at the following URL: https://gitlab.msu.edu/vermaaslab/LongBondElminiator (accessed on 31 December 2021), or via Zenodo [29]. We do note that the LongBondEliminator algorithm is stochastic in nature, since the result for individual simulation steps are not deterministic.

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
