# Peer review of "LongBondEliminator: A Molecular Simulation Tool to Remove Ring Penetrations in Biomolecular Simulation Systems"

_biomolecules, 2023, doi:10.3390/biom13010107_

Round 1

Reviewer 1 Report

Good paper. Although nothing is really new, the authors have developed a utility that many NAMD users will be happy to discover. A publication explaining the algorithms and applications is certainly proper in this case.

Notes:

When author reference program modules (e.g. colvars), a short (several words to a sentence) should be given to explain what the module actually does.

It would be useful (if possible) to visualize one of the examples with a set snapshots of fragments of structures to demonstrate the iteration that algorithm goes through creating of a good structure from a "pierced" structure. The current snapshots look a bit pointless.

typos: "Fig."  in some sentences, "Figure" in other sentences

Reviewer 2 Report

The authors have developed a longbondeliminator plugin that can be incorporated with VMD. The methods and results are discussed in detail, and the plugin can be used to sort out the systems' long bonds/piercing bonds. The authors took three different systems as examples. The manuscript can be accepted for publication after minor changes. 

Based on the results, the authors provide some additional information to further understand their results.

The study mainly dealt with minimizing the initial structure of the systems. How about the difference in energy scale between minimized energy with and without applying LBE? Could you comment on this?

The authors provide minor information on the computational time. They should give, how much computation time was taken for each system studied. What about increasing the system size? Will it take a long time prior to performing the actual simulation?  
